# [Re] Lifting 2D StyleGAN for 3D-Aware Face Generation

## Reproducibility Summary

*In this study, we present our results and experience during replicating the paper titled "Lifting 2D StyleGAN for 3D-Aware Face Generation" (1). This work proposes a model, called LiftedGAN, that disentangles the latent space of StyleGAN2 (2) into texture, shape, viewpoint, lighting components and utilizes those components to render novel synthetic images. This approach claims to enable the ability of manipulating viewpoint and lighting components separately without altering other features of the image. We have trained the proposed model in PyTorch (3), and have conducted all experiments presented in the original work. Thereafter, we have written the evaluation code from scratch. Our re-implementation enables us to better compare different models inferring on the same latent vector input. We were able to reproduce most of the results presented in the original paper both qualitatively and quantitatively.*

**Scope of Reproducibility**

In the scope of this study, we aim to reproduce all of the qualitative and quantitative results of LiftedGAN, including the ablation study, on FFHQ (4) and AFHQ Cat (5) datasets. Additionally, we further extend the experiments presented in the original work by testing the proposed approach on CelebA (6) dataset.

**Methodology**

We have adopted the source code for training from the author's repository. We have written the evaluation scripts from scratch in PyTorch to test the original and reproduced weights on the same latent vector. Our experiments have been completed on a single Nvidia Quadro RTX 6000 in 1 day for each, and it requires ~11GB GPU memory for training.

**Results**

We have achieved to reproduce the results qualitatively and quantitatively on a large scale. We also validated the generalization ability of the model by training and testing it on CelebA dataset. Although our experimental results are not identical with the original paper, they are consistent and validates the claims made by the original work.

**What was easy**

The paper is well-written. The main components of the LiftedGAN was open-source, and implemented in PyTorch, which facilitated our reproduction study.

**What was difficult**

3D evaluation and reconstruction scripts were not available in the official repository. Also, there were some missing implementation details to reproduce some results in the original work.

**Communication with original authors**

We were in contact with the authors since the beginning of the challenge. We could not achieve to reproduce 3D evaluation and reconstruction parts, fortunately, the authors swiftly answered our questions regarding the topic.

---

# 1 Introduction

The paper (1) proposes a framework that disentangles the latent space of a pre-trained StyleGAN2 (2) for 3D-aware face generation. The previous approaches are trained to generate random faces, thus they do not offer direct manipulation over the semantic attributes such as lighting or pose in the generated image. A number of studies exists that aims to manipulate the semantic attributes of the generated images directly (7; 8; 9; 10; 11). Although these feature manipulation methods have shown ability to generate faces with high visual quality under assigned poses, it is unclear whether other features such as identity are preserved when we change the pose parameters. In the paper (1), to overcome this problem, a pre-trained StyleGAN2 is distilled into a 3D-aware generator, which outputs the generated image with its viewpoints, light direction and 3D information.

The framework proposed in the original paper (1), namely LiftedGAN, is composed of five sub-networks that are responsible for light direction, viewpoint, foreground/background map, depth, and texture components. These sub-networks are than utilized to render a 2D face image. As the main claim of the paper, this method achieves to change the light direction and viewpoint without affecting the other important features such as texture and shape.

In this reproducibility report, we studied LiftedGAN for generating and manipulating human and cat faces. During this work, we have implemented the testing loops for running the experiments on the same randomly generated latent vectors. We have also trained both the StyleGAN2 and LiftedGAN models with different datasets from scratch. Furthermore, we present the results of the original work on different domains and compare the obtained results with the ones reported in the original paper. Finally, we report the important details about certain issues encountered during reproduction.

# 2 Scope of reproducibility

The main idea of the paper is to train a 3D generative network by distilling the knowledge in StyleGAN2 for building a 3D generator that disentangles the generation process into different 3D modules. Afterwards, those modules are utilized to render a 2D face image.

The proposed framework, namely LiftedGAN, claims to provide on-par performance to the state-of-the-art face generation methods in terms of Fréchet Inception Distance (FID) (12) score while providing the ability to change the viewpoint and light direction. To validate these claims, we try to investigate the following questions:

- Is the implementation details described in the paper and the provided code sufficient for replicating the quantitative results reported in the paper?
- Are the qualitative results visually-plausible?
- Could our replication obtain similar qualitative results compared to the reported qualitative results in the original paper?
- Could our replication obtain similar FID scores compared to the reported results in the original paper?
- How does the architecture perform when trained on other datasets (e.g. CelebA)?

# 3 Methodology

We have adopted the code for the architecture and the training loop from the official repository of the paper. Due to the nature of both StyleGAN2 and LiftedGAN, the framework samples a random latent vector from the latent space and uses that vector to generate a new face. This makes comparing the original and reproduced results not possible by using the original code, since the generated face is changed for each trial as we run the original test loop. To overcome this issue, we have written a modified version of the original testing loop that stores the randomly generated latent vector and provides it to different versions of the LiftedGAN model.

At this point, we found that the paper is well-written, and contains the details required to reproduce the most of the qualitative and some of the quantitative results. Since the official repository of the paper is publicly available, we mainly focused on reproducing the original experiments in a controlled manner and extending the experiments on different datasets to further validate the claims made by the original paper.

In this section, we introduce the implementation details of LiftedGAN, the points in the paper which were important for reproduction, hyperparameters we used, and our experimental setup.

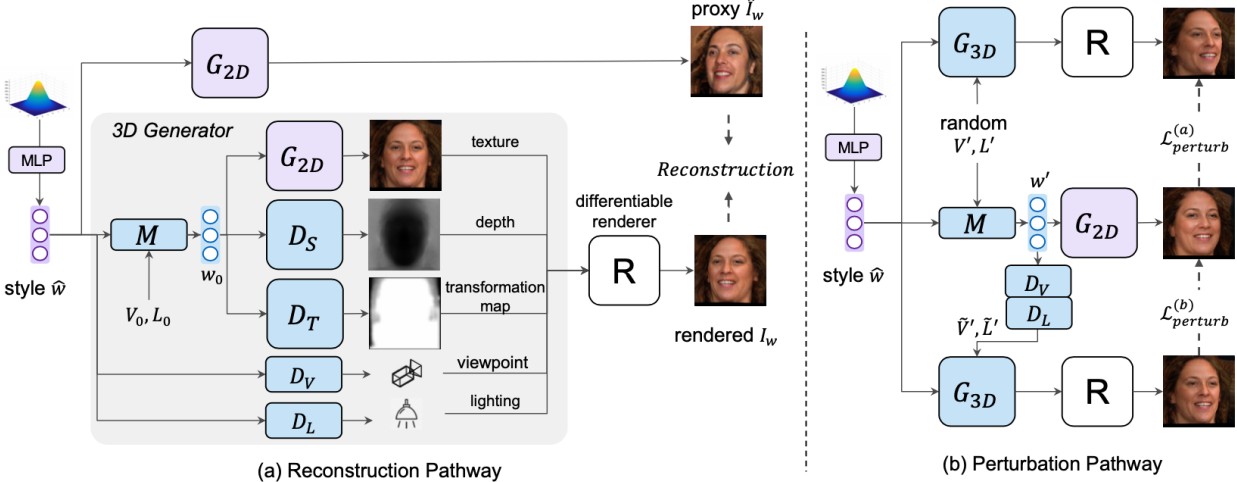

Figure 1: Overview of LiftedGAN architecture. The purple blocks indicate the modules from the pre-trained StyleGAN2, which are not updated during training. The blue blocks are the modules to be trained. Obtained from the original paper (1).

## 3.1 Model descriptions

The main idea of LiftedGAN is to train a 3D generative network by leveraging the knowledge in pre-trained StyleGAN2. The StyleGAN2 network is composed of two parts: a multi-layer perceptron (MLP) that maps a latent code $z \in Z$ to a style code $w \in W$, and a 2D generator $G_{2D}$ that synthesizes a face image from the style code $w$. LiftedGAN aims to build a 3D generator that disentangles the generation process of $G_{2D}$ into different 3D modules, including texture, shape, lighting and pose, which are then utilized to render a 2D face image. As shown in the Figure 1, the framework involves two pathways, which are the reconstruction pathway and style manipulation (*i.e.* perturbation) pathway.

### 3.1.1 3D Generator

As shown in Figure 1, the 3D generator, denoted as $G_{3D}$, is composed of five trainable sub-networks: $D_V$, $D_L$, $D_S$, $D_T$, $M$, a pre-trained StyleGAN2 $G_{2D}$ and a differentiable renderer $R$. $M$ is used as style manipulation network that transfers a style code $\hat{W}$ to a new style code with a specified lighting and viewpoint. This approach creates $w_0 = M(\hat{w}, L_0, V_0)$ thus, $G_{2D}(w_0)$ outputs a lighting and viewpoint neutralized face image. The rest of the sub-networks $D_V$, $D_L$, $D_S$, $D_T$ are responsible from the viewpoint, lighting, depth and shape representation, respectively. Finally, $R$ is used to output a rendered image $I_w = R(A, S, T, V, L)$ where $A$ is the face image with neutral viewpoint and lighting, $S$, $T$, $V$, $L$ are the depth, shape representation, desired viewpoint and desired lighting, respectively.

### 3.1.2 Loss Functions

As mentioned in Section 3.1, the framework has two pathways for face reconstruction and style manipulation. As shown in Figure 1, the reconstruction pathway uses L1 loss whereas the style manipulation pathway uses the perturbation loss. The overall reconstruction loss function consists of five objective functions, which are reconstruction loss $L_{rec}$, photometric flip loss $L_{flip}$, perturbation loss $L_{perturb}$, identity variance loss, $L_{idt}$ and albedo map loss $L_{reg_A}$. Overall loss function and its each component are defined below.

Reconstruction loss is defined as following:

$$L_{rec} = ||I_w - \hat{I}_w||_1 + \lambda_{perc} L_{perc}(I_w, \hat{I}_w) \tag{1}$$

where $L_{perc}$ refers to the perceptual loss (13) using a pre-trained VGG-16 network (14), $\hat{I}_w$ is the proxy image output by StyleGAN2 and $I_w$ is the image rendered by $R$. $L_{flip}$ has the same formulation as $L_{rec}$ except that it uses flipped albedo and shape maps during the rendering.

Perturbation loss is defined as following:

$$L_{LV_{cyc}} = ||\tilde{V}' - V'||^2 + ||\tilde{L}' - L'||^2$$

$$L_{perturb}^{(a)} = d(I_w', G_{2D}(w')) + \beta \frac{||w' - \mu_w||^2}{2\sigma_w^2}, L_{perturb}^{(b)} = d(R(A, S, T, V', L'), \hat{I}_w') + \lambda_{LV_{cyc}} L_{LV_{cyc}} \quad (2)$$

$$L_{perturb} = L_{perturb}^{(a)} + L_{perturb}^{(a)}$$

where $\hat{w}$ is a randomly sampled style code, $w'$ is the manipulated style code, $\hat{I}_{w'}$ represents the proxy image generated by the manipulated style code, $V'$ and $L'$ are the randomly sampled viewpoint and lighting vectors, $\tilde{V}' = D_V(w')$ and $\tilde{L}' = D_L(w')$. Also, $\mu_w$ is the empirical mean and $\sigma_w$ is the standard deviation of randomly generated style codes. $I_w'$ is the rotated and relighted face image output generated by $R(A, S, T, V', L')$. Identity variance loss component is defined as following:

$$L_{idt} = ||f(I_{w_0}) - f(I_w')||^2 \quad (3)$$

where $I_{w_0}$ is the texture map and $f$ is a pre-trained face recognition network. Albedo map loss component $L_{reg_A}$ is also defined as following:

$$L_{reg_A} = ||K_A||_* \quad (4)$$

where $K$ is the albedo matrix that is composed of filtered and vectorized albedo maps and $||.||_*$ denotes the nuclear norm. The overall loss function for the 3D generator used in the reconstruction pathway is as following:

$$L_{G_{3D}} = \lambda_{rec} L_{rec} + \lambda_{flip} L_{flip} + \lambda_{perturb} L_{perturb} + \lambda_{idt} L_{idt} + \lambda_{reg_A} L_{reg_A} \quad (5)$$

## 3.2 Hyper-parameters

The hyper-parameters used in the original work are mostly the objective function coefficients, and the default values mentioned in their paper are presented in Table 1. During our additional experiments on CelebA, we have followed the same settings that the authors used for FFHQ. We have also considered the batch size and learning rate as hyper-parameters, and they are set to $8$ and $1e-4$, respectively for all of our experiments.

Table 1: Objective function coefficients

| Dataset Name | $\lambda_{rec}$ | $\lambda_{perc}$ | $\lambda_{flip}$ | $\lambda_{perturb}$ | $\beta$ | $\lambda_{LVcyc}$ | $\lambda_{idt}$ | $\lambda_{regA}$ |
|---|---|---|---|---|---|---|---|---|
| FFHQ | 5.0 | 1.0 | 0.8 | 2.0 | 0.5 | 2.0 | 1.0 | 0.01 |
| AFHQ Cat | 5.0 | 1.0 | 0.8 | 2.0 | 4.0 | 0.0 | 1.0 | 0.005 |
| CelebA | 5.0 | 1.0 | 0.8 | 2.0 | 0.5 | 2.0 | 1.0 | 0.01 |

## 3.3 Datasets

Following the paper, we have conducted our experiments on two well-known datasets: FFHQ, AFHQ Cat. The original paper uses FFHQ for training the StyleGAN2, and the original LiftedGAN framework uses the generated data from the pre-trained StyleGAN2. Moreover, in the original work, AFHQ Cat is used to validate the performance of the architecture on a different domain. In addition to FFHQ, we have also conducted additional experiments on CelebA dataset to further validate the generalization ability of LiftedGAN. The details are provided in Table 1.

Table 2: Dataset details

| Dataset Name | Sample Size | Image Dimension | Training Dimension |
|---|---|---|---|
| FFHQ | 70,000 | 1024 ×1024 | 256 ×256 |
| AFHQ Cat | 5,000 | 512 ×512 | 256 ×256 |
| CelebA | 202,599 | 178 ×218 | 256 ×256 |

## 3.4 Experimental setup and code

In this study, we have followed the same protocol described in the original paper and the official repository for the FFHQ and AFHQ Cat experiments. For the additional experiments on CelebA, we have re-trained StyleGAN2 before training the LiftedGAN, which requires a pre-trained StyleGAN2.

126 We have used Fréchet Inception Distance (FID) score to measure the quantitative results, as in the original work.
127 Our implementation and the trained weights are open-sourced, and can be found at: `https://anonymous.4open.`
128 `science/r/lifting-2d-stylegan-for-3d-aware-face-generation-4B11`

## 3.5 Computational requirements

130 For this reproduction study, we have used 2 different machines to conduct our experiments. The first machine has an
131 AMD Ryzen 7 2700X CPU, 32 GB RAM and 2x Nvidia Quadro RTX 6000. The second one has Intel 3770K CPU, 8
132 GB RAM and 2x Nvidia GTX 1080.

133 StyleGAN2 trainings for our custom datasets have been conducted in our second machine, and take approximately 2-3
134 days to be completed, whereas LiftedGAN trainings have been conducted on our first machine, and completed in ~1
135 day. The experiments we conducted for reproducing this work do not require any other significant resources, but GPU
136 memory.

# 4 Results

138 We have conducted all experiments by following the descriptions given in the paper. We re-implemented the test
139 scripts that enables us to run two different models on a single latent vector. In general, we were able to reproduce the
140 quantitative and qualitative results on FFHQ and AFHQ Cat datasets. We extend the results of AFHQ Cat presented
141 in the original work by conducting the lighting and viewpoint (*i.e.* pitch) manipulation. Moreover, we extend the
142 experiments given in the original work by training the LiftedGAN from scratch and testing it on CelebA.

## 4.1 Results reproducing the original work

### 4.1.1 Qualitative results

145 As shown in Figure 2, we have achieved visually on-par face generation performance on FFHQ. Although there are
146 slight differences in our results compared to the results presented in the original work (*e.g.* the absence of glasses
147 in the second column and the first row), they do not reduce the face generation quality and the identical features for
148 all samples are mostly preserved. We provide more face generation examples for more extensive comparison in our
149 supplementary materials and the reproduction repository.

150 Figure 3 demonstrates the comparison of the viewpoint rotation between the outputs obtained by using the weights
151 given by the authors and the outputs reproduced by our work. At this point, we validate that LiftedGAN achieves to
152 change the viewpoints in the generated images without affecting the other visual features. Moreover, in Figure 4, we
153 show both qualitative results of the original work and our reproduction study on changing the direction of the light
154 source task on FFHQ dataset. We can state that LiftedGAN also achieves to change the direction of the light source in
155 generated images. In our study, we were able to reproduce these results.

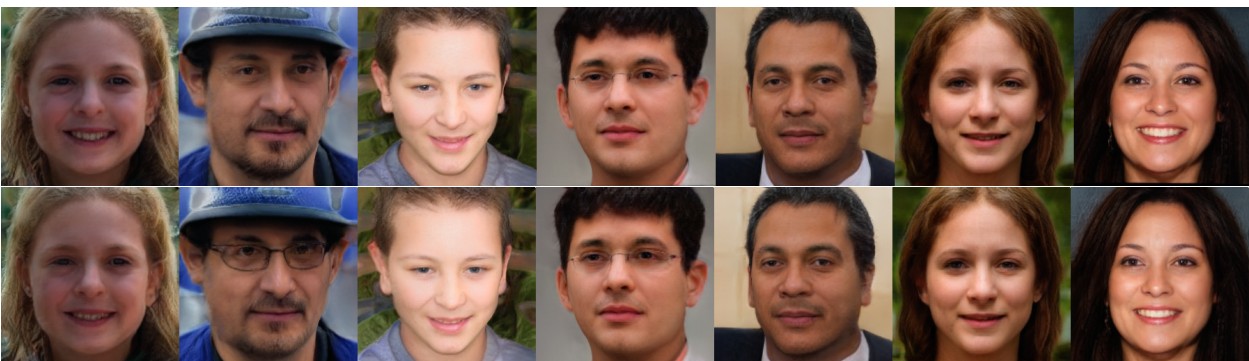

Figure 2: Face generation example on FFHQ. The first row is the results produced by the original weights, the second
row is the results produced by our reproduced weights.

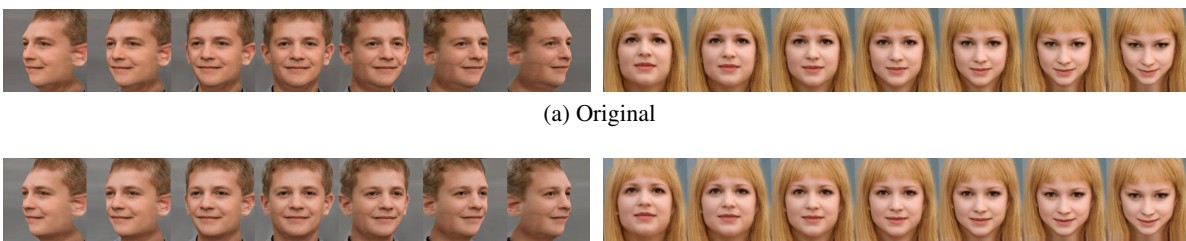

(a) Original

(b) Reproduced

Figure 3: The viewpoint rotation examples on FFHQ. The images on left demonstrate the changes in the yaw axis, while the images on right present the results of changing the pitch axis.

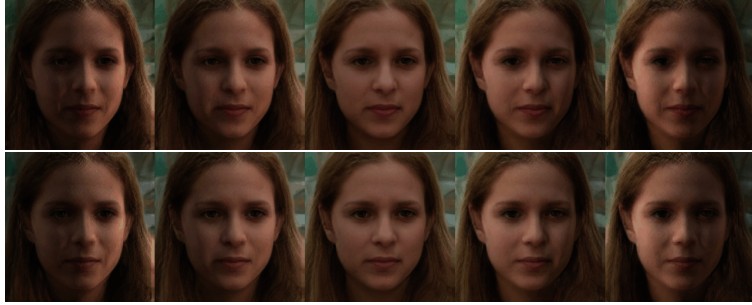

Figure 4: Changing the direction of the light source on FFHQ. The first row shows the results produced by the original weights, and the second row presents the results produced by our reproduced weights.

In the original work, the examples of face generation results between interpolated latent codes are demonstrated. The main claim in the paper is that LiftedGAN can achieve a smooth change between two disparate samples. To validate this claim, we have generated the face images by using the interpolated latent codes, and observed the effect of the viewpoint rotation strategy, as in the original work. Our reproduced weights can generate similar faces to the ones produced by the original weights with the same viewpoint rotations, as presented in Figure 6.

Qualitative results of the ablation study for our reproduction are shown in Figure 5. We also provide more visual examples for all these additional experiments in our supplementary materials and the reproduction repository.

### 4.1.2 Quantitative results

In this section, we present our quantitative results of this reproduction study in Table 3, and compare with the ones reported in the original work. The authors have conducted several ablation studies on FFHQ. Particularly, they remove symmetric reconstruction loss (*i.e., wo_flip*), perturbation loss (*i.e., wo_perturb*), identity regularization loss (*i.e., wo_idt*) and albedo consistency loss (*i.e., wo_rega*), respectively, to re-train their proposed architecture for further comparison. Our reproduced results have lower FID scores than the ones reported in the paper, as well as all ablation studies. As claimed in the original work, the model cannot produce visually-plausible and logically reasonable shapes for the generated faces, and this can be observed more dramatically in our reproduced results. Moreover, we additionally measure the performance of the proposed architecture and its variants on AFHQ, which is not reported in the original work. We obtain more similar quantitative results for the reproduction on AFHQ Cat dataset.

Table 3: Original and reproduced FID scores.

| Experiment Name | Dataset | FID (Reprod.) | FID (Orig.) |
|---|---|---|---|
| LiftedGAN wo_flip | FFHQ | 15.50 | 28.69 |
| LiftedGAN wo_perturb | FFHQ | 19.78 | 21.3 |
| LiftedGAN wo_idt | FFHQ | 24.44 | 30.63 |
| LiftedGAN wo_rega | FFHQ | 24.28 | 27.34 |
| LiftedGAN | FFHQ | 25.54 | 29.81 |

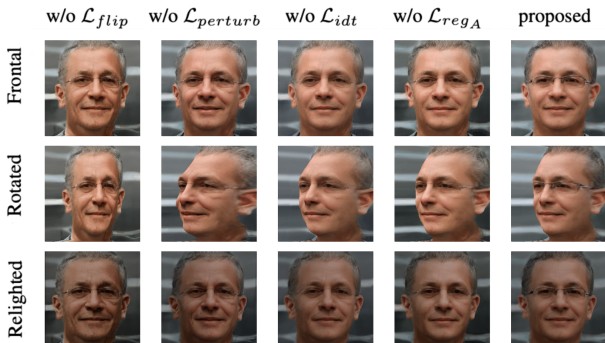

Figure 5: Reproduced qualitative results of ablation study.

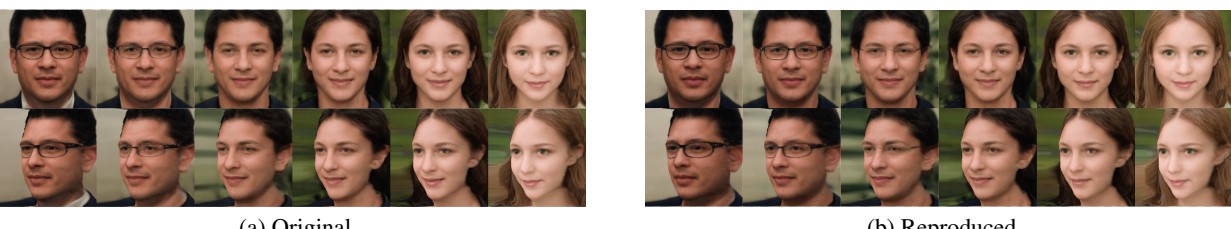

(a) Original                                          (b) Reproduced

Figure 6: Examples of using the interpolated latent codes for generating the rotated faces.

## 4.2   Results beyond the original work

### 4.2.1   Extended experiments on AFHQ

In the original work, a controlled generation strategy on cat heads has been followed in order to demonstrate that the framework is object-agnostic. However, this experiment is limited, and conducted on only the viewpoint manipulation on yaw axis. We present the visual results of our controlled generation on cat heads in Figure 7 (for the viewpoint manipulation in yaw and pitch axes) and in Figure 8 (for changing the light direction). At this point, we can validate that the framework is able to work well on different objects, not only human faces.

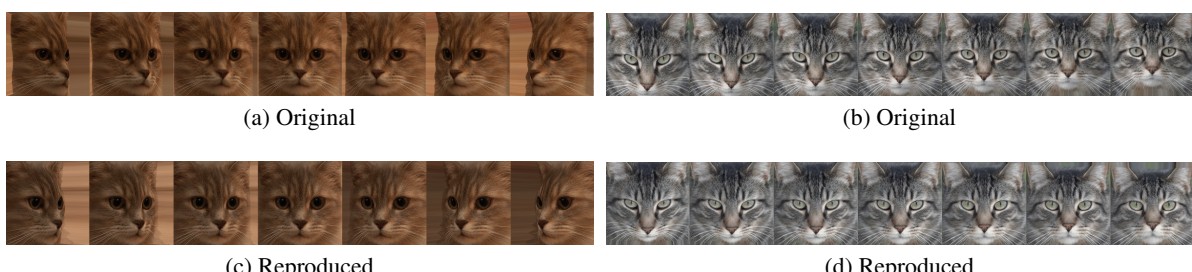

(a) Original                                          (b) Original

(c) Reproduced                                        (d) Reproduced

Figure 7: The viewpoint rotation examples on AFHQ Cat dataset. Left: Yaw axis, Right: Pitch axis.

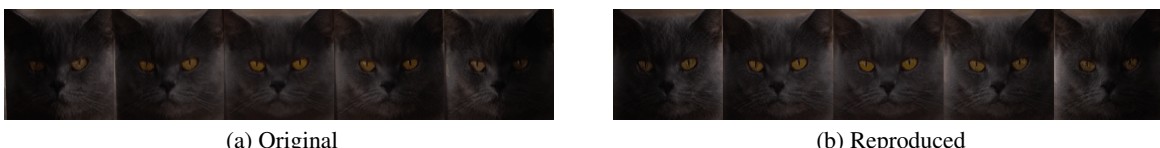

(a) Original                                          (b) Reproduced

Figure 8: Changing the direction of the light source on AFHQ Cat dataset.

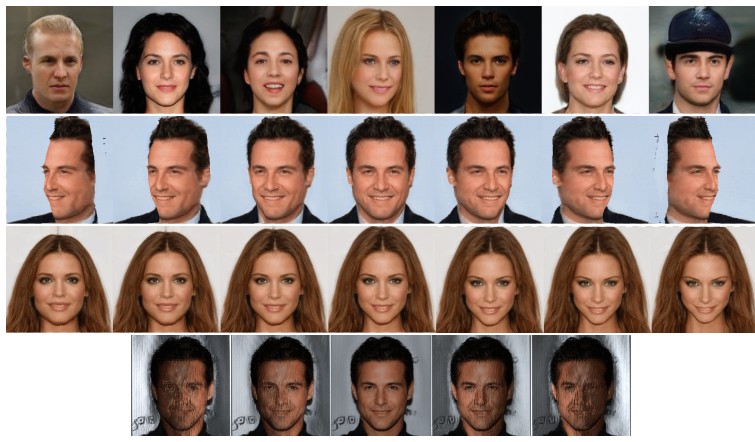

Figure 9: The qualitative results on CelebA dataset. Rows: (1) face generation, (2) rotation on yaw axis, (3) rotation on pitch axis, (4) the light direction.

### 4.2.2 The performance on CelebA

To extend the scope of the experiments in the original work, and validate the generalization ability of the architecture, we have re-trained the framework from scratch on CelebA. The visual results of this experiment can be seen in Figure 9. The main observations for this experiment are as follows: (1) the overall performance is similar to the one for FFHQ, (2) the outputs for the face generation is visually-plausible, (3) the viewpoint manipulation can be achieved on this dataset, (4) there are some visual artifacts in the outputs for the task of changing the light direction.

## 5 Discussion

We can clearly say that the paper reproduced was well-written. Although there are a few missing implementation details in the paper and a few missing evaluation scripts in the official repository, we were able to reproduce the results reported in the original work on a large scale. Overall, we were able to obtain similar qualitative results when compared to the original work. Our results are visually-plausible. The quantitative results do not exactly match with the reported results, but eventually not very far from them. In addition to these results, we demonstrate the reproduced results of the viewpoint rotation on yaw and pitch axes and changing the light direction tasks, the visual results of the ablation study and the task of generating interpolated and rotated faces. We extend the experiments on AFHQ Cat dataset, and also observe the performance of the proposed methodology on an additional dataset (*i.e.,* CelebA).

### 5.1 What was easy

The code was open-source, and implemented in PyTorch, hence adopting the training loop and model implementation facilitated our reproduction study. The provided pre-trained StyleGAN2 weights significantly reduced our required GPU hours for FFHQ experiments.

### 5.2 What was difficult

Since the 3D evaluation and reconstruction scripts are not available in the official repository and not described with enough detail in the original paper to reproduce it, we could not achieve to reproduce the results related to 3D reconstruction metric.

### 5.3 Communication with original authors

We were in contact with the authors since the beginning of the challenge. We could not succeed to reproduce the 3D reconstruction task, fortunately, they swiftly answered our questions, and provided more information for reproducing the task.

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
