# OpenReview forum: "[Re] Lifting 2D StyleGAN for 3D-Aware Face Generation"
_ML_Reproducibility_Challenge/2021/Fall — RC2021_

### Official Review · Reviewer_9V6K · 2022-03-08
**Lifting 2D StyleGAN for 3D-Aware Face Generation**

**Rating:** 7
**Confidence:** 4

**Review:**

This paper (re-paper) is well written, which reproduced most of the quantitative and qualitative results from the original paper: https://arxiv.org/pdf/2011.13126.pdf (ori-paper). The re-paper is scoped well in the beginning and accurately described the approach in the ori-paper including the loss functions and experimental setups. The "what's easy" and "what's difficult" sections are helpful for future researchers to re-investigate the ori-paper.

In addition to reproducing the ori-paper, additional hyper-parameters were considered and additional experiments were conducted on CelebA and AFHQ. The code of this re-paper is open-source.

---

### Official Review · Reviewer_vPo8 · 2022-03-31
**Good report but lacks in detail**

**Rating:** 4
**Confidence:** 4

**Review:**

The report reproduces the result of the CVPR 2021 paper titled "Lifting 2D StyleGAN for 3D-Aware Face Generation".

Scope of reproducibility: The study shows that the qualitative and quantitative results in the original paper are reproducible. Beyond that, the paper explores the performance of the model on a different dataset showing its generalizability.

Code: The authors have provided their code and they have reused the code of the authors mostly. They did not present results for which the code was not available.

Communication with original authors: Authors were in touch with the original authors all along.

Hyperparameter Search: The authors use the same hyperparameters as the original paper.

Ablation Study: The authors included the ablation study from the paper.

Discussion on results and Recommendations for reproducibility: The authors present a brief discussion of the results showing that the results were easy to reproduce. They claim that the paper was well written and code availability made it easy, but they do not provide more details beyond this. They vaguely mention that there were a few issues that the original authors helped resolve.

Results beyond the paper: The authors present a small extension to the authors' work on cat faces.

Overall organization and clarity: The paper does not provide much clarity or lacks in the following:
* The parts of the original paper that were hard to reproduce.
* The scope of this work seems limited given that the code was available and the original authors were able to clarify any missing information in the paper.

---

### Meta-Review · Area_Chair_rBWv · 2022-04-08

**Recommendation:** Accept
**Confidence:** 4

**Metareview:**

A good reproducibility study. The authors of the paper attempt to reproduce the results using code provided by the original authors of the paper and also present some additional results on celebA dataset verifying the generalizability of the proposed approach.

---

### Decision · Program_Chairs · 2022-04-09

**Decision:**

Accept

**Comment:**

Following the recommendation of reviewers and meta-reviewer, the paper is accepted for ML Reproducibility Challenge 2021, and will be published in the upcoming special edition of ReScience Journal.